# The Somatic Mutation Paradigm in Congenital Malformations: Hirschsprung Disease as a Model

**DOI:** 10.3390/ijms222212354

**Published:** 2021-11-16

**Authors:** Katherine C. MacKenzie, Rhiana Garritsen, Rajendra K. Chauhan, Yunia Sribudiani, Bianca M. de Graaf, Tim Rugenbrink, Rutger Brouwer, Wilfred F. J. van Ijcken, Ivo de Blaauw, Alice S. Brooks, Cornelius E. J. Sloots, Conny J. H. M. Meeuwsen, René M. Wijnen, Donald F. Newgreen, Alan J. Burns, Robert M. W. Hofstra, Maria M. Alves, Erwin Brosens

**Affiliations:** 1Department of Clinical Genetics, Erasmus University Medical Center-Sophia Children’s Hospital, 3000 CA Rotterdam, The Netherlands; k.c.mackenzie@outlook.com (K.C.M.); garritsenrhiana@hotmail.com (R.G.); rajchauhan65@gmail.com (R.K.C.); yungfbp@gmail.com (Y.S.); b.degraaf@erasmusmc.nl (B.M.d.G.); t.rugenbrink@erasmusmc.nl (T.R.); a.brooks@erasmusmc.nl (A.S.B.); Alan.Burns@takeda.com (A.J.B.); r.hofstra@erasmusmc.nl (R.M.W.H.); 2Department of Pediatric Surgery, Erasmus University Medical Center-Sophia Children’s Hospital, 3000 CA Rotterdam, The Netherlands; ivo.deblaauw@radboudumc.nl (I.d.B.); c.sloots@erasmusmc.nl (C.E.J.S.); c.meeussen@erasmusmc.nl (C.J.H.M.M.); r.wijnen@erasmusmc.nl (R.M.W.); 3Fluidigm Europe B.V., 1101 CM Amstelveen, The Netherlands; 4Department of Medical Biochemistry and Molecular Biology, Faculty of Medicine, Universitas of Padjadjaran, Bandung 45363, Indonesia; 5Department of Cell Biology & Center for Biomics, Erasmus University Medical Center Rotterdam, 3015 GD Rotterdam, The Netherlands; r.w.w.brouwer@erasmusmc.nl (R.B.); w.vanijcken@erasmusmc.nl (W.F.J.v.I.); 6Department of Paediatric Surgery, Amalia Children’s Hospital, Radboud University Medical Center, 6525 GA Nijmegen, The Netherlands; 7Department of Cell Biology, Murdoch Children’s Research Institute, Royal Children’s Hospital, Parkville, VIC 3052, Australia; don.newgreen@mcri.edu.au; 8Department of Stem Cells and Regenerative Medicine, UCL Great Ormond Street Institute of Child Health, London WC1N 1EH, UK; 9Takeda Pharmaceuticals, Cambridge, MA 02139, USA

**Keywords:** Enteric Nervous System, somatic mutation, Hirschsprung disease, missing heritability, gastrointestinal disease, motility disorder, developmental defects

## Abstract

Patients with Hirschsprung disease (HSCR) do not always receive a genetic diagnosis after routine screening in clinical practice. One of the reasons for this could be that the causal mutation is not present in the cell types that are usually tested—whole blood, dermal fibroblasts or saliva—but is only in the affected tissue. Such mutations are called somatic, and can occur in a given cell at any stage of development after conception. They will then be present in all subsequent daughter cells. Here, we investigated the presence of somatic mutations in HSCR patients. For this, whole-exome sequencing and copy number analysis were performed in DNA isolated from purified enteric neural crest cells (ENCCs) and blood or fibroblasts of the same patient. Variants identified were subsequently validated by Sanger sequencing. Several somatic variants were identified in all patients, but causative mutations for HSCR were not specifically identified in the ENCCs of these patients. Larger copy number variants were also not found to be specific to ENCCs. Therefore, we believe that somatic mutations are unlikely to be identified, if causative for HSCR. Here, we postulate various modes of development following the occurrence of a somatic mutation, to describe the challenges in detecting such mutations, and hypothesize how somatic mutations may contribute to ‘missing heritability’ in developmental defects.

## 1. Introduction

Congenital disorders can be caused by germline variants that, depending on the mode of inheritance, can be present in one or both parents of an affected individual. In the case of a dominant congenital disorder, the evolutionary pressure against disease-causing genetic variations is high, and for that reason mutations often occur de novo in a patient. These changes appear new in the patient and originate from a de novo mutation in the germ cells of one of the parents. An alternative route for the development of what seems to be an inherited dominant disease, is the occurrence of a somatic mutation(s) affecting a specific tissue(s). This process, somatic mosaicism, is a well-characterized phenomenon known to contribute to a number of diseases, most notably skin disorders such as, McCune–Albright syndrome and Darier–White disease [1,2]. Somatic mutations can be evenly distributed throughout an organism, be segmental or tissue specific and can affect somatic tissues, the germline or both. They can arise at any stage of development or adult life, may accumulate with age [3], and be caused by alterations of a normal to a mutant genotype and vice versa [4]. However, for a somatic mutation to play a significant role in tissue development and contribute to a congenital disorder, it is important that a threshold proportion of cells are affected. This situation most likely eventuates when the initiating mutation occurs at an extremely early stage of development [5], or when there is some form of clonal dominance such as, when the mutation brings selective advantage/survival to the mutated cells [6,7]. Cancers can be seen as examples of the latter [8].

In this study, we investigated the involvement of somatic mutations in a congenital neuropathy of the gastrointestinal tract (GIT) called Hirschsprung disease (HSCR). HSCR is a developmental disease characterized by inability of enteric neural crest cells (ENCCs) to colonize a variable length segment of the distal gut, which therefore lacks an Enteric Nervous System (ENS) [9]. HSCR is a genetic disorder and more than 20 genes have been identified that play a role in its development [10]. The REarranged during Transfection gene (*RET*), which codes for a receptor tyrosine kinase expressed by ENCCs, is the major gene for HSCR and mutations affecting its coding and non-coding regions have been described in the majority of patients [11,12]. However, the known genes only explain 30% of all HSCR cases [9]. This missing heritability seen in HSCR is a common feature of complex genetic disorders, and can partly be explained by low penetrant non-coding variants [13,14,15], as well as by combinations of both rare coding and modifying variants [16,17]. The occurrence of somatic mosaicism could also account for, or at least contribute to, some of the cases. However, in this case, mutations should preferably occur early in development to have an effect on the ENS. If they occur during development, these variants should result in a beneficial growth or survival advantage over the non-mutated cells, to colonize a substantial portion of the colon and result in a disease phenotype.

The theory that somatic variants may play a role in HSCR is not new, and has already been postulated in two and tested in three independent studies. The first study [18] investigated the presence of *RET* intronic variants [SNP1 (rs2506004) and SNP2 (rs2435357)] in a series of tissues (aganglionic, transition zone and ganglionic gut in distal to proximal order) collected from several HSCR patients. The authors observed that the aganglionic segments of HSCR patients tend to be homozygous (hemizygous) for the disease-associated variants, particularly in patients with long segment HSCR, whereas the ganglionic intestinal tissues of the same patient were found to be heterozygous. It was postulated that this was caused by a deletion of the wild-type allele [18]. A similar study was performed by a different group, but they were not able to find allele frequency differences for three *RET* intronic polymorphisms (rs2506030, rs2506004 and rs2435357) in DNA isolated from blood and colon of the same patient [19]. The third study described the existence of low-frequency *RET* somatic variants in different tissues from the same patient. In two patients, mutations were found in several tissues (blood, saliva and colon) at low frequencies and were absent in blood-derived DNA from the parents [20]. Although these studies claimed the involvement of *RET* somatic mutations in HSCR development, there are a number of caveats concerning the validity of the results. The first study stated that 100% of the alleles found in the aganglionic segment of HSCR patients, were mutated [18]. This implies that all examined cells had lost the wild-type allele. Since enteric ganglia are derived from ENCCs, which only constitute a minority of the cells of the gut, and no selection method was performed to enrich for the enteric neuronal population, it is likely that the authors analyzed not only ENCCs but also mucosal cells, connective tissue and smooth muscle cells. Since these cells derive from different germ layers, i.e., endoderm, mesoderm and ectoderm, respectively (Figure 1), the mutation identified must have occurred extremely early in development and should be present in all or most cells, in all tissues of the body. Therefore, identification of *RET* intronic mutations only in the aganglionic region is difficult to explain. The third study used a different approach to determine somatic mosaicism, but the questions raised are comparable, since *RET* mutations were identified in tissues derived from all three germ layers, making it difficult to discriminate between very early developmental stage somatic mutations, and de novo variants present as a germline mosaicism in one of the parents [21]. In light of these results, we believe that the involvement of somatic mutations in HSCR is still unclear. Here, we further investigate its existence by comparing whole-exome sequencing (WES) derived genotypes and copy number profiles of DNA isolated from ENCCs with either blood, or fibroblast samples collected from HSCR patients.

## 2. Results

### 2.1. Identification of Germline Mutations

For this study, 30 HSCR patients were collected, but only five underwent WES, due to the inability to grow ENCCS from the remaining patients. The quality of the data generated can be seen in Appendix A. No obvious deleterious germline mutation was identified in any of the known HSCR-associated genes, or in genes highly expressed in mouse ENCCs [16,23]. However, we found three protein-altering and ten synonymous (protein non-altering) variants in genes previously associated with HSCR (Appendix A). Twelve of these thirteen variants were believed to be benign/mild and only one was predicted as deleterious. This variant is located in the glial cell line-derived neurotrophic factor receptor alpha 1 (*GFRα-1*) and leads to loss of its starting codon. *GFRα-1* encodes for an extracellular protein that works as receptor for the glial cell line-derived neurotrophic factor (GDNF), and is required for RET activation. Although *GFRα-1* is considered to be a HSCR candidate gene, no pathogenic variants have been found in HSCR patients, making it difficult for us to assess the contribution of the variant identified to the phenotype. Considering that this variant would not be classified as causative in a diagnostic setting, we concluded that none of the patients analyzed had an obvious pathogenic germline mutation in a known HSCR gene, that could alone explain the ENS phenotype.

### 2.2. Identification of Somatic Variants

In order to detect the presence of somatic mosaicism in our cohort of patients, genetic variants identified in DNA from blood or fibroblasts, were compared to the ones found in DNA from ENCCs. Cell type-specific alternative alleles were identified (Table 1). Putative somatic variants were also found in four of the five patients analyzed (Table 2). Validation of these variants by Sanger sequencing failed to confirm their presence only in ENCCs, leading us to conclude that deleterious somatic changes could not be confirmed in the HSCR patients analyzed.

### 2.3. Analysis of Somatic Copy Number Changes

In total, 12 rare germline copy number variants (CNVs) were detected in our cohort of patients. Each patient holds at least one rare CNV (Appendix A). However, only the 1q25.3 gain (patient 1), the 14q24.1 gain (patient 2), the gain on 2p25.1 (patient 3) and the 6p22.3 loss (patient 5) contain genes, and can be classified as variants of unknown significance. None of the CNVs affected known HSCR genes (Appendix A). Somatic copy number changes were not detected in the ENCC populations. Additionally, inspection of the allele frequencies of germline and ENCC-derived profiles of these patients, did not reveal any significant differences.

## 3. Discussion

Establishing the involvement of somatic mosaicism in congenital malformations can explain disease occurrence in the absence of inherited or de novo coding mutations (present in the patient only). This is particularly important for counselling, as the recurrence risk for a tissue-specific somatic mutation is null, while a germline somatic mutation can still be present in the remaining germ cells and thus, be transmitted. In this study, we investigated if somatic variants substantially contribute to HSCR. We excluded pathogenic germline variants in known HSCR disease genes and specifically searched for the presence of somatic mutations in ENCCs isolated from colonic biopsies obtained from five HSCR patients. As somatic variation can originate at any stage of life, a fraction of all human cells are likely to carry a variant. Therefore, in order to be missed in a diagnostic setting, these variants should be present in ENCCs, but absent in blood cells or fibroblasts. WES of DNA isolated from purified ENCCs and blood or fibroblasts of these patients, resulted in a set of putative somatic mutations. However, none of these could be confirmed by traditional methods, suggesting that the variants identified are technical artefacts or noise. This result led us to conclude that in the patients analyzed, somatic variants do not play a role in HSCR development. As the number of patients was small we cannot exclude the involvement of these type of variants in HSCR. However, there are many reasons to assume that if somatic variants were to be involved in HSCR, they would likely be extremely difficult to detect. Here, we elaborate on this, and on the possible contribution of somatic mosaicism to the development of congenital disorders.

### 3.1. Detection of Somatic Mosaicism

In order to detect somatic mosaicism, analysis of multiple tissues within an individual is required. In some cases, the choice of tissue depends on the disease-associated phenotype; for example, in HSCR, the relevant cells to be investigated are ENCCs since an absence in their progeny in the distal gut, is the disease pathology. However, somatic mosaicism can also be searched for in the affected tissue only, using sensitive genotyping techniques such as single-nucleotide polymorphism (SNP) microarrays or next-generation sequencing (NGS) [4,24,25]. The downside of using such methods in a single tissue is that they will not prove the somatic nature of a presumed-somatic variant. This is due to the fact that discriminating technical artefacts from genuine somatic variants is challenging. Therefore, analysis of multiple tissues within an individual is preferable. However, somatic variants might be present in low frequencies in tissues from multiple germ layers, including the affected tissue. If this is the case, they would have arisen at or before the epiblast stage of embryogenesis, and be virtually indistinguishable from a mutation that arose in the germ cells of one of the parents. Technical distinction between these types of mosaicism is challenging. If alternative alleles are present in the DNA of both blood and ENCCs, the allele frequency will likely be high and fall within the normal range (between 20–70%), meaning that they will appear as de novo heterozygous variants. To circumvent these issues, we searched for somatic variants (alternative alleles) present only in the affected tissue, by comparing WES data from two cell types of the same patient. Although we were unable to identify real somatic variants in the patients analyzed, we believe that this is not dependent of our experimental design, but simply due to the fact that somatic mutations do not contribute to HSCR development in these patients. One could argue that this is due to sensitivity of Sanger sequencing. In our hands, the sensitivity for a known variant is approximately 10%, and the minimal variant quality for validation of somatic changes was set accordingly. Of course we cannot exclude the possibility that some mutations might be missed if the real allele frequency is lower than 10%, but this is unlikely to be the case in all validated variants. A more likely explanation for the lack of validation is that we detected artefacts in our WES analysis. This ‘sequencing noise’ is inherent to WES and is influenced by DNA quality, which explains the higher number of detected SNP in the ENCCs of patient 4, the sample with the lowest DNA quality. Although the number of variants looks high, it is in fact extremely low compared to the total number of true variants identified, as the concordance rate between cell types at 20X coverage is 98.6–99.6%. Inclusion of parental information would have helped to reduce the noise substantially, as it would allow identification of de novo variant(s), and better discriminate artefacts from true somatic mosaicism. However, this was only possible for one patient (patient 5) due to a lack of parental DNA for the other cases.

### 3.2. ENS-Specific Somatic Changes in DNA Copy Number

Somatic mosaic CNVs have been described previously [26]. We know that the brain is especially sensitive to them [27,28], and hypothesized that this also happens in ENCCs. The detection limit for new changes is close to 10% and we can detect allele-specific differences between cell types, close to 20% [29]. As expected, we identified several germline changes in DNA copy number, most of them were known polymorphisms and not related to HSCR. However, no germline changes differed in allele frequency in the ENCCs, nor did we find new ENCC-specific alterations. In conclusion, we could not identify somatic differences in ENCCs, neither small variants nor larger DNA copy number variations in the patients analyzed.

### 3.3. Would ENCCs with a Somatic Variant Remain to Be Sampled?

To answer this question, we have to consider two parameters: the development of the ENS and different models that could represent the effect of a somatic variant on the subsequent distribution of mutated cells. The ENS is mainly formed from the vagal neural crest, with a small contribution from the sacral neural crest. Cells of the neural crest migrate from the folding neural tube to enter the oral end of the developing gut (Figure 2).

These ENCCs migrate rostro-caudally within the gut mesoderm, following chemo-attractive and proliferative signals along the gut tube. As the gut is concomitantly growing and elongating, the developing ENCCs are highly migratory, with individual cells migrating locally in all directions, but with a caudal net direction. The leading wavefront sets the tracks for other cells to follow, determining the position of the later ganglionic network [30].This complex development of the ENS can be disrupted by germline defects, which affect all ENS precursor cells. A somatic mutation would only affect the cells containing the mutation and thus, its effect would be dependent on when and in which lineage the variant arose, as well as which proteins and functions are affected directly and indirectly. There are countless possible models that could represent the effect of a somatic mutation in HSCR, but we believe that they can be grouped into three broad categories: no selective effect, selective advantage, and selective disadvantage (Figure 3).

The first model, no selective effect, can be explained by assuming that the mutation will always be passed on to the daughter cells without any selection effect against the mutated cell (Figure 3A). This model is somewhat comparable to a PCR reaction. When a mutation is introduced in the first cycle, its detection after 30 cycles depends on the number of copies of that specific allele on the starting DNA fraction. Detecting such mutations can only be done by sequencing single products for instance, via cloning of the PCR product. In this model, the phenotype could only be seen if the variant was to occur very early in development, when there are few total cells [5] or if another mechanism for restricting clonal variance occurs, such as trans-mesenteric migration [31,32]. This model is the simplest, and is likely not representative for progenitors of the ENS, as we believe that without a selective effect the population of mutated cells would not result in HSCR.

Mutations that result in a selective advantage for the cell are also unlikely to be present in HSCR, as the disease is characterized by a loss of cells in the ENS (Figure 3B). Selective variants are a hallmark of cancer, and are seen in overgrowth syndromes [4]. However, if selective mutations were to occur, they should be detectable with a high allele frequency [33], and this is not the case for HSCR patients. Interestingly, a stochastic model involving “superstar” cells has been proposed for normal ENS development. This model was proven using computer modelling and grafting experiments, and showed that the eventual ENS is formed by just a few cells, named superstars [31]. In HSCR, we believe that the opposite is more likely to occur, namely that the variant causes a selective disadvantage for the affected cells, leaving them unable to migrate, proliferate or differentiate. As a consequence, a reduced number of functional cells would be available to colonize the gut. If using this model we expect aganglionosis to occur due to somatic changes of vagal neural crest cells, we know that the sacral neural crest will also be unable to colonize the distal portion of colon [34]. Based on this evidence, we hypothesize that if an unfavorable variant occurred very early in development or in a superstar cell, the chances of detecting such variant in the ganglionic biopsies sampled as part of this study, are incredibly low, as the affected cells are unlikely to have reached the end of the GIT. Similarly, if the mutation inhibited ENCC specification to ganglionic cell lineages, these cells would not exist. These outcomes might, of course, also result in low allele frequencies due to the low number of cells that would reach the distal colon. Therefore, more proximal regions of the GIT would have to be examined in order to find higher allele frequencies.

## 4. Materials and Methods

### 4.1. Patients

Thirty HSCR patients undergoing colonic pull-through surgery in the Paediatric Surgery department of the Erasmus Medical Center, Sophia Children’s Hospital, Rotterdam and the Radboud Medical Center, Amalia Children’s Hospital, Nijmegen, were included in this study. All patients were operated in their first year of life. An overview of the clinical characteristics of the patients is given in Appendix A.

### 4.2. Sample Collection

Full-thickness colonic biopsies from patients were obtained from the most proximal region of removed colon from pull-through surgeries, and confirmed as ganglionic. Colonic biopsies were washed with sterile PBS and connective tissue was removed. The tissue was dissected and dissociated in 200 U/mL Collagenase IV (Gibco, Thermo Fisher Scientific, Waltham, MA, USA) for 1 h at 37 °C. EDTA blood and a skin biopsy (2 mm) were also collected. Fibroblasts were used as source of DNA, when insufficient DNA was isolated from blood.

### 4.3. Cell Culture and Fluorescence-Activated Cell Sorting

Skin biopsies were dissected and plated in Ham’s F-10 nutrient mix (Gibco, Thermo Fisher Scientific, Waltham, MA, USA) supplemented with 15% foetal calf serum (FCS) and 1% penicillin/streptomycin. Medium was refreshed every 2–3 days and once confluent, cells were split at a ratio of 1:3, using TrypLE Express (Gibco, Thermo Fisher Scientific, Waltham, MA, USA) according to the manufacturer’s instructions. Dissociated colon cells were strained with a 100 µm cell strainer (Falcon, Corning, Glendale, AZ, USA) to yield a single-cell suspension. Cells were plated on fibronectin-coated plates (Invitrogen, Thermo Fisher Scientific, Waltham, MA, USA) to form neurosphere-like bodies, as previously described [35]. Medium was refreshed every 2–3 days and cells were expanded in vitro for 1–4 weeks. Cultures were split using Accutase (Sigma Aldrich, Burlington, MA, USA). To isolate ENCCs from the mixed population, the cell culture was dissociated, strained with a 100 µm cell strainer, washed in PBS containing 10% FCS and stained with an antibody against p75^NTR^, a neural crest marker, conjugated with phycoerythrin (1:100, ab157333, Abcam, Waltham, MA, USA). Cells were sorted using a BD FACSAria™ III (BD Biosciences, Franklin Lakes, NJ, USA), and snap-frozen in liquid nitrogen for DNA isolation.

### 4.4. Amplicon-Based WES

Genomic DNA was isolated from peripheral blood using standard methods. Genomic DNA was isolated from fibroblasts and ENCCs with the QIAamp DNA Micro kit (Qiagen, Venlo, The Netherlands), according to the manufacturer’s instructions. DNA libraries for WES were constructed using 250 ng of germline DNA and 250 ng of DNA isolated from ENCCs captured with the Haloplex exome target enrichment kit (Agilent Technologies, Santa Clara, CA, USA). Captured fragments were sequenced [paired-end 101 base pair (bp) read length] on the Illumina HiSeq2500sequencers (Illumina, San Diego, CA, USA). Raw sequence data were processed using the Nimbus Suite [36]. Reads were aligned to the hg19 reference sequence, and alternative and reference alleles were counted per genomic position.

### 4.5. Data Analysis and Selection of Somatic Variants

WES data were analyzed to exclude (likely) pathogenic variants in known HSCR disease genes (Appendix A). When determining cell type-specific somatic variations (ENCC or germline), the alternate allele had to be present (1) at least, 5 times in 2 amplicons, (2) at least in 10% of reads, and (3) be absent in the other cell type analyzed. The minimal coverage per base used in the analysis was 20× in both cell types. Protein-altering variants were only considered when the minor allele frequency in GnomAD exome and/or GnomAD genome v2.1 (http://gnomad.broadinstitute.org/, accessed on 16 November 2018), was below 0.001.

The following criteria were subsequently used to prioritize putative somatic variations on deleteriousness and involvement of genes in ENS development: changes in putative loss of function or predicted to affect splicing [37] in intolerant genes [37]; changes with a CADDv1.4 score of 15 or higher (http://cadd.gs.washington.edu/home, accessed on 16 November 2018) [38]; and/or changes with a predicted deleteriousness in the best performing prediction tools from the first three clusters previously described [37,38,39,40,41,42,43,44,45,46,47]. mRNA expression in the developing ENS was inspected using publicly available mouse data sets (Gene expression omnibus: GSE34208 and GSE111307). Prioritized genes were the ones with a mouse orthologue differentially expressed between (a) E14.5 intestine or ENS cells [23], or (b) E11.5 and E15.5 ENS, progenitors or intestine [48]. We also used available in-house RNA sequencing data from human intestine at embryonic weeks 12, 14 and 16 [49]. Enteric nervous genes expression prioritization has been described previously [50].

### 4.6. Validation of Putative Mosaic Differences

Top ranking ENCC-specific variants (based on either quality or deleteriousness) were validated using Sanger sequencing, as previously described [23]. The primers are available on request.

### 4.7. Analysis of Somatic Copy Number Changes

Germline DNA of all five patients was inspected for the presence of rare deleterious copy number changes. ENCC-derived DNA was also inspected for patient 1, 2 and 3. An insufficient amount of ENCC DNA was available for patients 4 and 5. Copy number changes were determined using either the HumanOmni5-4_v1.1 beadchip (patient 1 and 2) or the Infinium Global Screening Array-24 v1.0 (patient 3, 4 and 5) (Illumina Inc., San Diego, CA, USA). All protocols and procedures were performed as previously described [51]. All profiles were inspected visually with the Biodiscovery Nexus CN8.0 (Biodiscovery Inc., Hawthorne, CA, USA), with special focus on allele frequency differences between copy number changes present in germline and ENCC-derived DNA.

## 5. Conclusions

In this study, we investigated the involvement of somatic mosaicism in HSCR as a model to study the occurrence of such event in congenital malformations. Based on the evidence collected here, we are inclined to suggest that it is unlikely that somatic mutations are involved, as these variants would automatically result in a selective disadvantage for the affected cell. As a consequence, the affected cells are likely not present at the end of the GIT, making the chances of detecting such variants in ganglionic biopsies sampled as part of this study, incredibly low. Similarly, if the mutation inhibited ENCC specification to ganglionic cell lineages, these cells would not exist. However, these outcomes could, of course, also result in low allele frequencies due to the low number of cells that would reach the distal colon. This fact, together with our small patient cohort, makes it difficult for us to completely rule out the involvement of such mutations in HSCR. What we can only say is that we did not identify somatic mutations in the five HSCR patients included in this study. We believe that with new technologies emerging, the use of smaller concentrations of DNA will be possible to analyze cells obtained from suction biopsies from multiple regions along the GIT. In this way, regions where higher allele frequencies are present could be investigated. Moreover, single-cell sequencing techniques might provide a powerful alternative to mass sequencing modalities, to give a true picture of the extent of mosaicism present in various tissues [52]. These advances, together with the appropriate separation of cell lineages, selection of sequencing modality and filtering of variants, will make it easier to determine whether somatic mutations play a role in congenital GIT disorders and if their occurrence rate can explain at least, part of the missing heritability seen for most of these disorders.

## Figures and Tables

**Figure 1 ijms-22-12354-f001:**
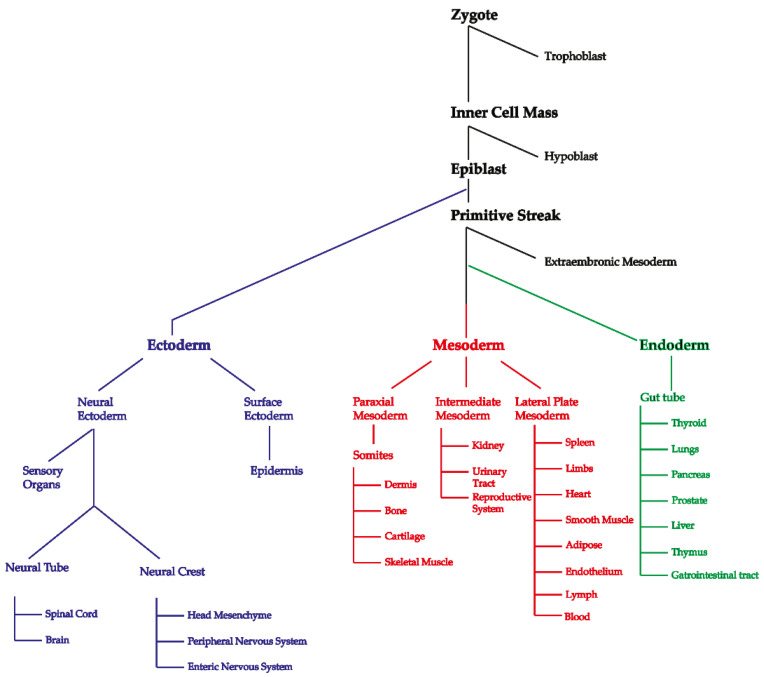
Developmental lineage tree highlighting where each of the three germ layers branch is from and what structures and organs they form or contribute to. Blood samples would be mesodermal, saliva samples would contain leukocytes of mesodermal origin and epithelial cells of ectodermal origin from the surface ectoderm [22], and unsorted gut samples would contain derivatives of all three germ layers, with ectodermal neural crest making up for the minority of cells. This figure was adapted from LifeMap (http://discovery.lifemapsc.com (accessed on 16 November 2018)).

**Figure 2 ijms-22-12354-f002:**
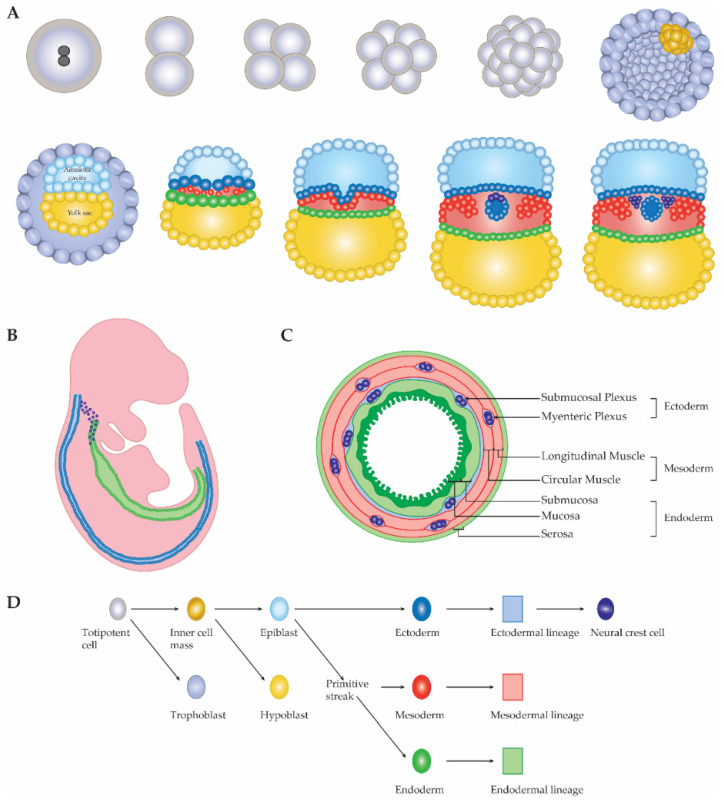
Schematic representation of early embryonic development. (**A**) First divisions and differentiations of the developing embryo, depicting the formation of the three germ layers and the neural crest cells. (**B**) The highly proliferative and migratory ectodermal neural crest, begins to enter the endodermal gut tube at week 4 of human gestation. Some of these cells will also contribute to the neural cells of the lungs and the pancreas. (**C**) Cross section of the colon to highlight contribution from all three germ layers, with the lowest contribution being from ectoderm/neural crest. (**D**) Legend of cell types and their origins.

**Figure 3 ijms-22-12354-f003:**
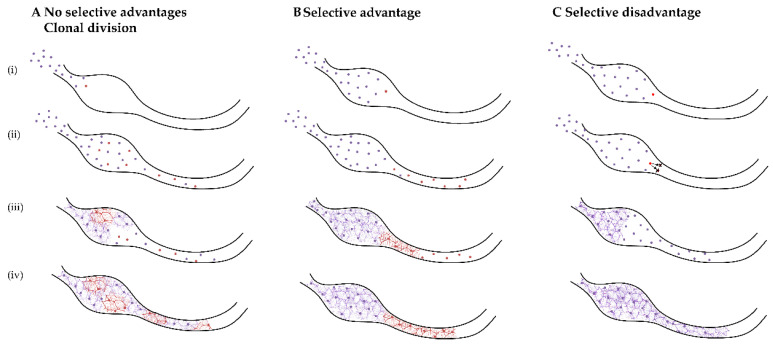
Proposed models of somatic mutations leading to HSCR. Each model shows four developmental stages, from initial ENCC migration into the gut tube (i), further migration and proliferation (ii), initial formation of ganglia (iii), to final colonization and ganglia formation (iv). (**A**) In the non-selective clonal growth and division model, the variant brings no competitive advantage or disadvantage, during migration or proliferation. However, this may lead to decreased differentiation or maturation capacity, for example. (**B**) In the selective advantage model, the variant confers a proliferative or migratory advantage to the cells, allowing them to out-compete the wild-type cells. However, due to early differentiation, altered subtype specification or inability to follow extracellular signals, the cells are unable to reach the distal portion of gut. (**C**) In the selective disadvantage model, the variant affects migration of ENS precursors, or the viability of the cells. As a consequence, the speed of the migratory wavefront is affected, leading to failure of ENCCs to colonize the distal colon. Apoptosis can also occur at some stage of migration or differentiation, resulting in decreased total cell count, that is insufficient to fully colonize the length of the gut tube. This is the most likely model that would result in a HSCR phenotype. Purple dots represent ENCCs with “normal” genotype; red dots represent ENCCs that have acquired a somatic variant.

**Table 1 ijms-22-12354-t001:** Number of variants identified in each patient, passing subsequent quality and prioritization criteria.

Patient	ENCC Only	Blood Only	PPAE	PPAB	VAPE	Validated @
1	50	43	8	11	5	0
2	16	28	2	4	1	0
3	25	33	0	1	0	0
4 ^#^	96	178	17	11	15	0 ^$$^
5	29	35	2	1	2	0

^#^ see filtering steps describing the selection criteria for the best somatic candidate mutations; @ see variant prioritization criteria, ^$$^ No ENCC DNA available for Sanger sequencing validation, only exclusion of the variant in fibroblast-derived DNA. ENCC only: All putative somatic mutations in ENCC only, Blood only: All putative somatic mutations in blood only, PPAE: protein-altering alleles in ENCC only, PPAB: protein-altering alleles in blood only, VAPE: Putative somatic mutations in ENCC meeting the prioritization criteria, Validated: number of somatic mutations confirmed with Sanger sequencing.

**Table 2 ijms-22-12354-t002:** Variants identified in DNA isolated from ENCCs of each patient, passing quality criteria and prioritization based on predicted deleteriousness, expression pattern and sensitivity for a gene to rare variation (conserved coding regions).

Patient	Gene	cDNA	Type	dbSNP	Class	GnomADe	GnomADg	MisZ	pLI	FE	ME
1	*FMN2*	c.162delC	FD		VUS	0.000000	0.000000	1.42	0.99	no	yes
1	*YWHAE*	c.G142A	M		VUS	0.000000	0.000000	3.25	0.96	yes	na
1	*YWHAE*	c.T116C	M		VUS	0.000000	0.000000	3.25	0.96	yes	na
1	*PHAX*	c.C379T	PS		VUS	0.000000	0.000000	–0.51	0.00	yes	na
1	*POR*	c.T1231C	M		VUS	0.000000	0.000000	−0.54	0.00	yes	na
2	*DEPDC1*	c.T1459A	M		VUS	0.000000	0.000000	−0.32	0.00	yes	na
4	*F5*	c.A1867G	M		VUS	0.000000	0.000000	−1.30	0.00	no	na
4	*PHRF1*	c.G1075A	M	rs551874512	VUS	0.000033	0.000032	−1.36	0.95	yes	na
4	*MYBPC3*	c.C482A	M		VUS	0.000000	0.000000	0.69	0.00	no	na
4	*PACS1*	c.G1069A	M	rs750459659	VUS	0.000041	0.000032	4.32	1.00	yes	na
4	*OAS3*	c.C1390T	M	rs750291946	VUS	0.000012	0.000000	−0.60	0.00	yes	na
4	*MAN2A2*	c.G478A	M	rs374688808	VUS	0.000012	0.000032	1.28	0.00	yes	yes
4	*SNF8*	c.G578A	M	rs775611332	VUS	0.000025	0.000000	0.97	0.29	yes	yes
4	*MED15*	c.C730A	M		VUS	0.000000	0.000000	2.50	0.96	yes	na
4	*IQCF5*	c.C283T	M	rs772101978	VUS	0.000100	0.000000	−1.59	0.43	no	na
4	*TMEM165*	c.C782A	M		VUS	0.000000	0.000000	1.83	0.94	yes	na
4	*NOTCH4*	c.G1118A	M	rs745883985	VUS	0.000033	0.000032	2.45	0.00	yes	na
4	*DPPA5*	c.G214A	M		VUS	0.000000	0.000000	1.64	0.00	no	na
4	*SLC22A1*	c.C523T	M	rs768905186	VUS	0.000004	0.000000	−0.28	0.00	no	na
4	*MGAM2*	c.G3015T	M		VUS	0.000000	0.000000			na	na
4	*IKBKB*	c.G809A	M	rs200841053	VUS	0.000024	0.000032	2.90	1.00	yes	na
5	*PCDH15*	c.G139A	M		VUS	0.000000	0.000000	−3.27	0.00	no	yes
5	*ZNF592*	c.C3433A	M		VUS	0.000000	0.000000	1.10	0.95	yes	na

More detailed information is available in Appendix A. No variants passed the quality criteria for putative somatic mutations in patient 3. FD: Frameshift deletion, M: Missense, VUS: Variant of Unknown Significance, LD: Likely Deleterious, PS: premature stop codon; FE: Human Fetal Expression EW12-EW16 in logCP, ME: Mouse ENS expression E11-15.5, GnomAd v2.1: exome and genome population frequencies, PLI: probability of loss of function intolerant, MisZ: missense Z score.

## Data Availability

All relevant data are within the manuscript and/or its Appendix A. Our ethics committee does not allow sharing of individual patient or control genotype information in the public domain.

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
