# Peer review of "The Somatic Mutation Paradigm in Congenital Malformations: Hirschsprung Disease as a Model"

_ijms, 2021, doi:10.3390/ijms222212354_

Round 1
Reviewer 1 Report
MacKenzie et al. described a genetic screening for somatic mutations on ENCCs and blood/fibroblasts of patients with Hirschsprung disease using amplicon-based whole exome sequencing. The major culprit of the current study is the small number of patients (n=5) being examined, which is inconclusive for the overall contribution of somatic mutations to the disease. While high coverage is needed for the investigation of somatic mutations, the mean coverage is not mentioned in the manuscript.
Comments:
1) Although the hypothesis given in the discussion is interesting, it is speculative and is not tightly linked to the current findings of the study.
2) The number of samples being examined is not enough to conclude the contribution of somatic mutations to the disease. While most of the previous studies focused on the major disease gene, increasing the sample size while focusing on the major gene(s) can provide more solid evidence for the absence of major somatic mutations in patients.
3) No mean coverage was given in Supplementary Table S1. Coverage of 20X is not enough to evaluate somatic mutations unless the mutation is present in a large proportion of cells. In addition, read information seems to be missing for ENCCs of patient 1.
4) Proportion/number of reads supporting the possible somatic alleles should be given.
Minor comments:
1) Most of the "Rare germline variants in HSCR disease genes" are not rare.
2) For table 2, there is no indication of which cell types are the alternative alleles identified from. In addition, it was mentioned in the footnote that "More detailed information is available in supplementary table S2"; however, the supplementary table on possible somatic mutations is not present.
3) Page 10, line 378-379: WES data was analyzed to exclude (likely-) pathogenic variants in known HSCR disease genes (Supplementary table S3); however, Supplementary table 3 is related to rare CNVs.
Author Response
Reviewer 1
1) Although the hypothesis given in the discussion is interesting, it is speculative and is not tightly linked to the current findings of the study.
- We thank the reviewer for his/her comment. To link our ideas in a more clear way, we have modified slightly the conclusion of our study. Please see page 11. “ In this study, we investigated the involvement of somatic mosaicism in HSCR, as a model to study the occurrence of such event in congenital malformations. Based on the evidence collected here, we are inclined to suggest that it is unlikely that somatic mutations are involved, as these variants would automatically result in a selective disadvantage for the affected cell. As a consequence, the affected cells are likely not present at the end of the GIT, making the chances of detecting such variants in the ganglionic biopsies sampled as part of this study, incredibly low. Similarly, if the mutation inhibited ENCC specification to ganglionic cell lineages, these cells would not exist. However, these outcomes could, of course, also result in low allele frequencies due to the low number of cells that would reach the distal colon. This fact, together with our small patient cohort, makes it difficult for us to completely rule out the involvement of such mutations in HSCR. What we can only say is that we did not identify somatic mutations in the five HSCR patients included in this study. We believe that with new technologies emerging, the use of smaller concentrations of DNA will be possible to analyze cells obtained from suction biopsies from multiple regions along the GIT. In this way, regions where higher allele frequencies are present could be investigated. Moreover, single-cell sequencing techniques might provide a powerful alternative to mass sequencing modalities, to give a true picture of the extent of mosaicism present in various tissues [53].These advances, together with the appropriate separation of cell lineages, selection of sequencing modality and filtering of variants, will make it easier to determine whether somatic mutations play a role in congenital GIT disorders and if their occurrence rate can explain at least, part of the missing-heritability seen for most of these disorders.
2) The number of samples being examined is not enough to conclude the contribution of somatic mutations to the disease. While most of the previous studies focused on the major disease gene, increasing the sample size while focusing on the major gene(s) can provide more solid evidence for the absence of major somatic mutations in patients.”
- We understand the reviewer’s comment and agree that the number of patients tested in this study is small. However, we have included in total, 30 HSCR patients. We have isolated ENCCs from the transition zone of these patients, and tried to culture the cells to isolate enough DNA. We assumed that if we were able to have a pure ENCC population, a somatic mutation could be easily detected. However, the ENCCs of most patients did not survive in culture. In retrospect, this fact fits our model as these cells are likely “ unhealthy” and therefore, unable to survive in culture after neuronal differentiation. Regarding the fact that we did not focus in the major disease gene, this was simply because we know that only 30% of all HSCR cases are explained by mutations in the known HSCR genes. Thus, for this project we decided to use an exome-wide approach to investigate the presence of somatic mutations that might be present in new modifier genes (in addition to a high predisposing haplotype burden or pathogenic germline variant). In this way, we were simply trying to increase our possibility to find somatic mutations.
3) No mean coverage was given in Supplementary Table S1. Coverage of 20X is not enough to evaluate somatic mutations unless the mutation is present in a large proportion of cells. In addition, read information seems to be missing for ENCCs of patient 1.
- We appreciate the reviewer’s comment, and have now added the mean coverage of the potential disease genes to Supplementary S1. We are aware that deep sequencing is required when somatic mutations are being identified from a tissue sample, composed by different cell types. However, for our study, we specifically looked for the presence of somatic variants in DNA collected from ENCCs. If a somatic mutation would be present in these cells, it would have a relatively high allelic frequency and thus, be detectable at modest sequencing depth (Brosens et al., 2021). Additionally, we decided to use an amplicon-based sequencing technique, which can detect variants that are present in two amplicons, as well as in the forward and reverse strands. Such technique, allows for a more precise discrimination of false positives and true variants, that if present, should be detectable by Sanger sequencing if the location of the putative mutation is known.
4) Proportion/number of reads supporting the possible somatic alleles should be given.
- As suggested by the reviewer, we have now added three extra supplementary tables, called Supplementary table S3A, S3B and S3C. These tables contain the requested information (S3A. Coverage over all genes per sample, S3B. Coverage over known HSCR disease genes, and S3C. Additional information of the putative somatic mosaic variants identified).
Minor comments:
1) Most of the "Rare germline variants in HSCR disease genes" are not rare.
- As suggested by the reviewer we deleted the word “rare” from the title of Supplementary Table S2.
2) For table 2, there is no indication of which cell types are the alternative alleles identified from. In addition, it was mentioned in the footnote that "More detailed information is available in supplementary table S2"; however, the supplementary table on possible somatic mutations is not present.
- The reviewer is totally right. The variants described in Table 2 are the ones identified in the DNA isolated from ENCCs of the five patients analyzed. We have now added this information to the title of the table. We have also changed the sentence "More detailed information is available in supplementary table S2" to "More detailed information is available in supplementary table S3C"
3) Page 10, line 378-379: WES data was analyzed to exclude (likely-) pathogenic variants in known HSCR disease genes (Supplementary table S3); however, Supplementary table 3 is related to rare CNVs.
- We thank the reviewer for pointing this mistake out. We were actually referring to Supplementary table S2 and not S3. We have now corrected this mistake in the text.
Reviewer 2 Report
No mention of getting informed consent or approval of the study by the hospital research ethics committee.
The manuscript “The somatic mutation paradigm in congenital malformations: Hirschsprung Disease as a model” is a well-written report of a small study looking for evidence of ENCC-specific mutations in children with Hirschsprung disease with a nice theoretical overview of the developmental issues involved.
I have a few concerns/suggestions:
- The level of genetic sophistication expected from the reader is rather uneven. The introduction is written for someone with limited genetics background but the results section assumes the reader is comfortable with genetic terms like “synonymous variants.” I suggest the authors review the whole manuscript to make sure they are not slipping into more advanced genetics language without explaining the terms to the reader.
- Regarding Figure 3 and the related discussion about mutational advantage and migration, the modeling does not seem to consider non-cell autonomous effects in ENS precursor migration. Could there be mutations that specifically affect contact inhibition of locomotion? A somatic mutation in a subset early migrating ENS precursors could produce a selective survival disadvantage reducing the total number of ENS precursors. This could affect the speed of the migration wavefront and lead to failure to colonize the distal colon – while non-mutant cells enter the colon and the mutant cells remain in the proximal gut.
- More information is needed regarding the selection of ENCC cells. A post-natal section of gut was used to select P75 positive cells. Do you know what proportion of the cells selected had a neural or glial phenotype vs other cells. What investigations were done regarding the quality of the selection process? This would be important to your discussion of the sensitivity of the sequencing to detect mutations.
Author Response
Reviewer 2:
1) The level of genetic sophistication expected from the reader is rather uneven. The introduction is written for someone with limited genetics background but the results section assumes the reader is comfortable with genetic terms like “synonymous variants.” I suggest the authors review the whole manuscript to make sure they are not slipping into more advanced genetics language without explaining the terms to the reader.
- We appreciate the reviewer’s comments and have revised the manuscript to make sure the genetic language is kept to the same level throughout.
2) Regarding Figure 3 and the related discussion about mutational advantage and migration, the modeling does not seem to consider non-cell autonomous effects in ENS precursor migration. Could there be mutations that specifically affect contact inhibition of locomotion? A somatic mutation in a subset early migrating ENS precursors could produce a selective survival disadvantage reducing the total number of ENS precursors. This could affect the speed of the migration wavefront and lead to failure to colonize the distal colon – while non-mutant cells enter the colon and the mutant cells remain in the proximal gut.
- The point raised by the reviewer is very important. We tried to address the migratory affect with the selective disadvantage model. However, we noticed that we haven’t describe it clearly. We have now mentioned this in the text, so our theories can be better understood. Please see page 8, line 304-307: “In the selective disadvantage model, the variant affects migration of ENS precursors, or the viability of the cells. As a consequence, the speed of the migratory wavefront is affected, leading to failure of ENCCs to colonize the distal colon. Apoptosis can also occur at some stage of migration or differentiation, resulting in decreased total cell count, that is insufficient to fully colonise the length of the gut tube.”.
3) More information is needed regarding the selection of ENCC cells. A post-natal section of gut was used to select P75 positive cells. Do you know what proportion of the cells selected had a neural or glial phenotype vs other cells. What investigations were done regarding the quality of the selection process? This would be important to your discussion of the sensitivity of the sequencing to detect mutations.
- The reviewer brings up a very interesting point. Unfortunately, we do not know the precise cellular composition of the P75 selected population. However, we only did cell sorting as a final step of our procedure. We first cultured the single cell suspension obtained after dissociation of the post-natal section, so we could obtained neurospheres. After that, we dissociated the neuropsheres to obtain a monolayer culture of neuronal cells, and only then we sorted the cells. Furthermore, we have also performed immunofluorescence stainings on these cultures using specific neuronal markers, such as Tuj1, before we started this project. As we wanted to make sure that we were looking indeed at a neuronal population.
4) No mention of getting informed consent or approval of the study by the hospital research ethics committee.
- We appreciate the reviewer’s comment and are aware of the importance of such information. We kindly ask the reviewer to check page 11, under “Informed Consent Statement” and “Institutional Review Board Statement”, respectively.
Round 2
Reviewer 1 Report
Major comments:
1) The authors stated that the number of samples under study is 30, which is sufficient to draw the conclusion for insignificant contribution of somatic mutations; however, only five patients were present in the manuscript? Could you please clarify the sample size?
Minor comments:
1) Supplementary Table should be ordered according to the appearance in the main manuscript.
2) Still mean coverage is not presented in Table 2.
Author Response
1) The authors stated that the number of samples under study is 30, which is sufficient to draw the conclusion for insignificant contribution of somatic mutations; however, only five patients were present in the manuscript? Could you please clarify the sample size?
- Indeed, we included in this study 30 HSCR patients. However, we were only able to successfully established ENCCs culture for five of them. That is the reason why we decided to focus only on these five patients, as they were the ones from which we isolated DNA from ENCCs and blood/fibroblasts and could therefore, look for the present of somatic variants. To make the number of patients included in this study more clear, we added a sentence to the Results section, page 4, line 150-151: “For this study, 30 HSCR patients were collected, but only five underwent WES, due to the inability to grow ENCCS from the remaining patients. “. We also changed the number of patients included in this study in the material and methods section, page 9, line 345-348: “Thirty HSCR patients undergoing colonic pull-through surgery in the Paediatric Surgery department of the Erasmus Medical Center, Sophia Children’s Hospital, Rotterdam and the Radboud Medical Center, Amalia Children’s Hospital, Nijmegen, were included in this study.”.
Minor comments:
1) Supplementary Table should be ordered according to the appearance in the main manuscript.
- As suggested by the reviewer we changed the order of the supplementary tables presented.
2) Still mean coverage is not presented in Table 2.
- The reviewer is right and we apologize for this. We meant to refer to Supplementary table S3, specifically to S3A for the average coverage over the entire gene, and S3C as having the actual depth of the variant location. We have now corrected this, and the information requested by the reviewer can be found in supplementary tables S2A and S2C.
Reviewer 2 Report
The authors have adequately addressed my previous concerns.
Author Response
We thank the reviewer for his comments and time.
Round 3
Reviewer 1 Report
The authors have addressed my concerns. I have no further comments.